# DISTILLING TRANSFORMERS INTO NEURAL NETS FOR FEW-SHOT TABULAR CLASSIFICATION

## ABSTRACT

Transformer-based models have shown promising performance on tabular data compared to their classical counterparts such as neural networks and Gradient Boosted Decision Trees (GBDTs) in scenarios with limited training data. They utilize their pre-trained knowledge to adapt to new domains, achieving commendable performance with only a few training examples, also called the few-shot regime. However, the performance gain in the few-shot regime comes at the expense of significantly increased complexity and number of parameters. To circumvent this trade-off, we introduce TabDistill, a new strategy to distill the pre-trained knowledge in complex transformer-based models into simpler neural networks for effectively classifying tabular data. Our framework yields the best of both worlds: being parameter-efficient while performing well with limited training data. The distilled neural networks surpass classical baselines such as regular neural networks, XGBoost and logistic regression under equal training data, and in some cases, even the original transformer-based models that they were distilled from.

## 1 INTRODUCTION

Tabular data plays a central role in many high-stakes applications, ranging from finance and healthcare, to manufacturing and weather prediction (Shwartz-Ziv & Armon, 2022; van Breugel & van der Schaar, 2024). However, the scarcity of labeled data can limit the application of machine learning in some of these domains, e.g., some diseases are extremely rare, or certain natural phenomena occur once in centuries (Hegselmann et al., 2023; Nam et al., 2023). In financial applications, annotating data can be expensive, and suffer from issues such as subjectivity, mislabeling, lack of consensus, and also data imbalances where only the data of accepted applicants may be available but not the rejected group (Crook & Banasik, 2004). Thus, tabular classification models that perform well under limited training data, also called the few-shot regime, are of immense interest.

Recently, transformer-based models have been shown to surpass classical approaches such as neural networks and Gradient Boosted Decision Trees (GBDTs) in the few-shot regime when the number of training examples is significantly small (Hollmann et al., 2023; Hegselmann et al., 2023; Jayawardhana et al., 2025). While GBDTs such as XGBoost (Chen & Guestrin, 2016), CatBoost (Prokhorenkova et al., 2018), and LightGBM (Ke et al., 2017) have long been the state-of-the-art for tabular classification when there is sufficient labeled data for training (Shwartz-Ziv & Armon, 2022; Grinsztajn et al., 2022), transformer-based models instead exploit their pre-trained knowledge to achieve improved performance in the few-shot regime. However, the performance gain in the few-shot regime comes at the expense of efficiency. Transformer-based models are extremely complex (millions or billions of parameters) in comparison to traditional neural networks and GBDTs, requiring massive compute, energy, and time during inference. To be able to cater to applications across varying levels of infrastructure, it is usually desirable that the deployed models are parameter-efficient and scalable. In this work, our key question is: *Can we achieve the best of both worlds, i.e., being parameter-efficient while also performing well with limited training data?*

Toward answering this question, we propose TabDistill, a framework to distill the classification capabilities of pre-trained transformers into neural networks for few-shot tabular classification. We draw inspiration from the image domain where transformer-based models have been found to be good hypernetworks for generating neural networks to implicitly represent images (Chen & Wang, 2022; Gu & Yeung-Levy, 2025). *Succinctly, TabDistill incorporates the pre-trained knowledge of a*

*transformer-based model (the base model) into a neural network by fine-tuning the transformer to infer its weights.* We assume that the base model contains an informative intermediate representation (for example, the encoder output of TabPFN (Hollmann et al., 2023) or encoder-decoder type language models such as BERT (Devlin et al., 2019), BART (Lewis et al., 2019), BigScience T0 series (Sanh et al., 2021) etc.). TabDistill learns a linear map for projecting the intermediate representation provided by the base model into the parameter space of the neural network, by fine-tuning using the cross-entropy loss of the resultant classifier. We employ a novel permutation-based training technique to avoid overfitting the model to the extremely small number of training examples.

Our experiments span over four tabular datasets and two base models. We compare the TabDistill framework with 5 baselines, including 3 classical models and the 2 base models. Experimental results indicate that the neural network distilled using the proposed framework exceeds the classical baselines in performance, particularly in the very-few-shot regime (when the number of training examples is less than 10). Interestingly, under some settings, the distilled neural network exceeds the performance of the base model which it was distilled from. In summary, our contributions can be listed as follows:

- **Propose TabDistill, a novel framework to distill transformers into neural networks.** We introduce a way to extract the performance of transformers into a much more efficient Multi-Layer Perceptron (MLP). Accordingly, the framework has additional advantages of the resulting model being differentiable and more easily explainable.

- **Instantiate the framework with two transformer-based models.** We instantiate the distillation framework with two transformer-based models Bigscience T0pp (Sanh et al., 2021) and the more recent TabPFN (Hollmann et al., 2023), which have $\sim$11B and $\sim$11M parameters respectively. We distill these base models into significantly simpler neural networks with $\sim$1000 parameters.

- **Experimental validation.** We conduct experiments on four tabular datasets (Bank (Moro et al., 2014), Blood (Yeh, 2008), Calhousing (Pace & Barry, 1997) and Income (Kohavi, 1996)) and five baselines (MLP, logistic regression, XGBoost, and the two base models TabPFN and T0pp). The distilled MLP surpasses the classical baselines in the few-shot regime under equal training data, and in some cases, even the original transformer-based models that they were extracted from.

## 1.1 RELATED WORKS

**Classical algorithms for tabular data.** Despite the success of deep learning in various other domains, classical machine learning algorithms such as logistic regression and GBDT methods such as XGBoost (Chen & Guestrin, 2016), LightGBM (Ke et al., 2017) and CatBoost (Prokhorenkova et al., 2018) have been dominating the domain of tabular datasets Shwartz-Ziv & Armon (2022). While highlighting the lack of a proper benchmark and a standard way of tuning hyperparameters for a fair comparison, Grinsztajn et al. (2022) point out that the deep learning models struggle on tabular datasets mostly due to difficulties in learning irregular patterns of the target function. Multiple works have focused on overcoming such difficulties and adapting neural networks for tabular datasets (See Gorishniy et al. (2024); Arik & Pfister (2021); Popov et al. (2019) and references therein). However, given the fact that these classical models are trained from scratch for a given dataset, their performance degrades significantly in the few-shot regime (Hegselmann et al., 2023).

**Transformer-based models for tabular data.** Transformer-based models have seen promising performance gains within the tabular data domain. A multitude of works employ the transformer as a way to model complex interactions between features of a tabular dataset. SAINT (Somepalli et al., 2022) uses an attention mechanism across rows as well as columns to better learn the structures within data. It also incorporates a self-supervised pre-training method for situations where the labels are scarce. Hollmann et al. (2023) trains a transformer from scratch on a massive collection of synthetic tabular datasets sampled from a causal mechanism. The trained transformer TabPFN can then be used to predict new tabular tasks with no additional training. In an attempt to leverage the pre-trained knowledge of a Large Language Model (LLM) for tabular data classification, Hegselmann et al. (2023) fine-tune models from BigScience T0 series (Sanh et al., 2021) to achieve remarkable performance in the few-shot regime. Jayawardhana et al. (2025) proposes PFN-Boost and LLM-Boost techniques where a pre-trained transformer is incorporated as the initial weak classifier of a GBDT ensemble. However, the performance gain of these methods is offset by the increased complexity and resource consumption particularly during inference. Moreover, the increased complexity

causes difficulty in assessing reliability, for instance, model multiplicity (Hamman et al., 2025). Our method focuses on mitigating these limitations by distilling the transformer into an MLP.

**Meta-Learning and hypernetworks.** Meta-learning refers to the process of learning to generalize to unseen tasks by observing few examples corresponding to each task (Vilalta & Drissi, 2002). Transformers are known to be good at meta learning, particularly in the form of in-context learning (Kirsch et al., 2022). Hypernetworks are closely related meta-learning, in the sense that they predict parameters for other machine learning models by observing a few samples from the task at hand. Transformers have been used as hypernetworks in computer vision applications, specifically for generating implicit neural representations (Chen & Wang, 2022; Gu & Yeung-Levy, 2025). Chen & Wang (2022) uses a transformer trained from scratch to predict weights of a neural network which represents an image or a 3D scene. Gu & Yeung-Levy (2025) exploits the pre-trained knowledge of a transformer-based foundation model for the same task. Both these works append additional placeholder tokens to the input for predicting the neural network weights. In contrast, our framework directly maps the embedding space to neural network parameters.

## 2 TABDISTILL: DISTILLING TRANSFORMERS INTO NEURAL NETWORKS

Here, we first discuss our proposed TabDistill framework along with the training procedure and possible methods for hyperparameter tuning. We then elaborate on two example instantiations of the framework using two popular transformer-based models for few-shot tabular data classification, namely, TabPFN (Hollmann et al., 2023) and TabLLM (Hegselmann et al., 2023).

### 2.1 NOTATION AND PROBLEM SETUP

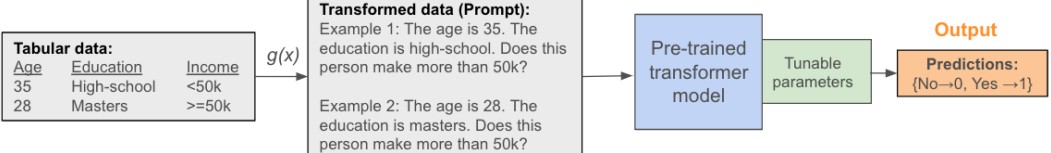

(a) The TabLLM framework. Tabular data is first converted to a natural language string using a serialization technique (denoted by $g(x)$). The serialized text is given as the input to the LLM and a prediction is directly generated as the output. Fine-tuning the LLM can improve classification performance.

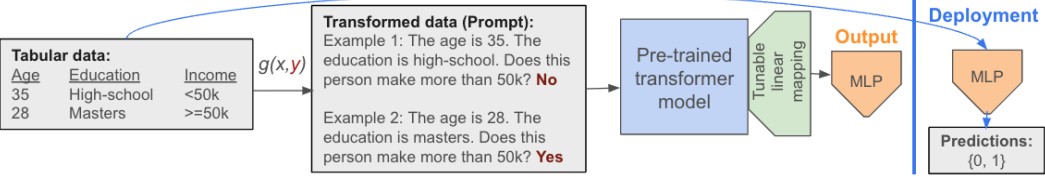

(b) Our TabDistill framework. Similar to TabLLM, the serialized text is given as the input to the LLM. However, in contrast to TabLLM, an MLP is generated as the **final** output of the transformer model. Only this MLP is deployed for making predictions on real-world data. Fine-tuning the LLM gives an improved MLP.

Figure 1: Comparison of TabLLM and TabDistill frameworks. The tunable parameters which are fine-tuned during training in each framework are depicted in green. The example dataset contains `Age` and `Education` as features. The target is to predict whether the `Income` is $>= 50k$ or not.

Let $\mathcal{D}_N = \{(x_n, y_n), x_n \in \mathcal{X}, y_n \in \{0, 1\}, n = 1, \ldots, N\}$ be a small tabular dataset for binary classification with $d$ features (usually pre-processed, e.g., categorical features are one-hot encoded) and $N$ datapoints ($N \sim 10$). Our focus is on transformer-based models capable of classifying instances $x \in \mathcal{X}$ of tabular datasets. To this end, LLMs have been adapted as few-shot classifiers through parameter-efficient fine-tuning (Hegselmann et al., 2023). To perform classification using an LLM, the tabular data instance $x$ must first be transformed into a natural language string (denoted by $s \in \mathcal{S}$ where $\mathcal{S}$ is the space of all possible strings within a given length). Hegselmann et al. (2023) studies a wide array of techniques for converting rows of a tabular dataset into text, known

as "serialization techniques". These techniques include using a fixed text template such as "The <column_name> is <value>" and using a list template of the form "<column_name>: <value>." We denote such a serialization by $g(x) : \mathcal{X} \to \mathcal{S}$.

The text output of the LLM can be converted to a binary class prediction using a similar technique (for example, Yes $\to 1$, and No $\to 0$). We abstract out this mapping and denote the transformer model by $f(s) : \mathcal{S} \to \{0, 1\}$. Note that with this setup, a meaningful classification can be carried out by predicting $\hat{y} = f(g(x))$. For brevity, let $g(x, y)$ represent a similar transform applied to both the features and the label together, and $g(\mathcal{D}_N)$ represent the concatenation of all the strings $g(x_n, y_n)$ corresponding to each $(x_n, y_n) \in \mathcal{D}_N$. See Figure 1 for an example application of a text template. It is worth noting that TabPFN (Hollmann et al., 2023) takes the tabular feature values themselves as the input and hence, $\mathcal{S} = \mathcal{X}$ and $g(x)$ in this case is the identity function, i.e., $g(x) = x$.

*Our goal is to use the pre-trained knowledge of the complex transformer-based model $f$ to generate a much simpler MLP $h_\theta(x) : \mathcal{X} \to \{0, 1\}$ with parameters $\theta \in \Theta$ that can classify $x \in \mathcal{X}$.* The intuition is that the pre-trained knowledge of $f$ will assist in generating $h_\theta$ effectively in a few-shot setting (i.e., when $N$ is very small). We consider the complex model $f(s)$ as consisting of two major components: an encoder $f_E(s) : \mathcal{S} \to \mathcal{Z}$ and a decoder $f_D(z) : \mathcal{Z} \to \{0, 1\}$, where $\mathcal{Z}$ is an embedding space. This is the case for the transformer-based models used in TabLLM (Hegselmann et al., 2023) and TabPFN (Hollmann et al., 2023) as well as popular LLMs such as BERT (Devlin et al., 2019), BART (Lewis et al., 2019), and T5 (Raffel et al., 2020).

The MLP $h_\theta(x)$ has the following architecture. Let $\texttt{ReLU}(u)$ denote the ReLU activation function (Glorot et al., 2011). With the hyperparameters $R$ and $L$ denoting the number of layers and the width of the hidden layers, respectively, $h_\theta(x)$ is defined as

$$h_\theta(x) = \texttt{ReLU}\left(W_R \texttt{ReLU}\left(\cdots \texttt{ReLU}\left(W_2 \texttt{ReLU}\left(W_1 x + b_1\right) + b_2\right) \cdots\right) + b_R\right) \qquad (1)$$

where $W_i$ and $b_i$ $(i = 1, \ldots, R)$ are the weights and the biases of each linear layer. The parameter $\theta$ denotes the combination of all such weights and biases, i.e., $\theta = (W_1, b_1, W_2, b_2, \ldots, W_R, b_R)$ and hence, $\dim(\Theta)$ is equal to the total number of tunable parameters in $h_\theta$ determined by $d$, $R$ and $L$. The first matrix $W_1 \in \mathbb{R}^{L \times d}$, where $d$ is the input dimension. All the intermediate layers have $W_i \in \mathbb{R}^{L \times L}$ for $i = 2, \ldots, R - 1$ and the final layer has $W_R \in \mathbb{R}^{L \times 2}$ for binary classification. For all $i = 1, \ldots, R$, $b_i \in \mathbb{R}$. The output logits $h_\theta(x)$ can be normalized by applying a Softmax function $\sigma(\cdot)$ to get the final class probability predictions. If desired, one can also choose different dimensions for each weight matrix rather than a fixed $L$.

## 2.2 OUR PROPOSED TABDISTILL FRAMEWORK

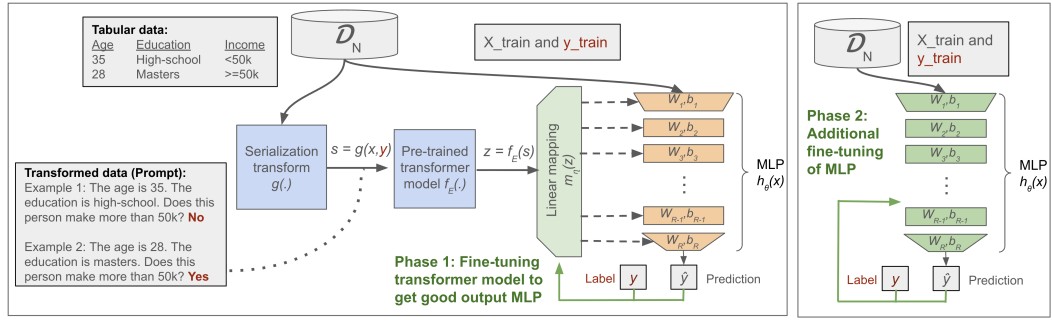

Figure 2: TabDistill framework. In Phase 1 (left), the tunable parameters of the transformer model (the linear mapping $m_\eta(z)$) is fine-tuned, as depicted in green. The resultant output MLP $h_\theta$ is depicted in amber. When T0pp is used as the base model $f$, a text serialization $g(x, y)$ is applied as shown in the figure. When TabPFN is used as the base model, $g(x, y)$ becomes the identity function. In Phase 2 (right), the MLP may be further fine-tuned if desired, as depicted in green.

**Phase 1: Fine-tuning the base transformer model.** The distillation is achieved by using the encoder $f_E$ of the complex model for inferring the weights of the MLP $h_\theta$. We learn a linear mapping $m_\eta(z) : \mathcal{Z} \to \Theta$ parameterized by $\eta$ such that $\theta = m_\eta(f_E(g(\mathcal{D}_N)))$ results in a useful classifier $h_\theta$. We use a simple normalized linear layer as the mapping function, defined as $m_\eta(z) =$

`LayerNorm`$(Az + b)$, where $A \in \mathbb{R}^{\dim(\Theta) \times \dim(\mathcal{Z})}$, $b \in \mathbb{R}^{\dim(\Theta)}$ and $\eta = (A, b)$. During Phase 1, the fine-tuning loss $\mathcal{L}(\eta; \mathcal{D}_N)$ is computed as follows:

1. Create a combined serialized input $g(\mathcal{D}_N)$ for the complex model $f$ using the training data $\mathcal{D}_N$
2. Prompt the complex model encoder to get the embeddings $z = f_E(g(\mathcal{D}_N))$
3. Infer the parameters $\theta = m_\eta(z)$ for the MLP $h_\theta$
4. Compute the loss $\mathcal{L}(\eta; \mathcal{D}_N)$ as the cross entropy loss of the classifier $h_\theta$

Ultimately, the fine-tuning loss function can be written as

$$\mathcal{L}(\eta; \mathcal{D}_N) = \sum_{n=1}^{N} y_n \log\Big(\sigma(h_\theta(x_n))[[1]]\Big) + (1 - y_n) \log\Big(\sigma(h_\theta(x_n))[[0]]\Big) \tag{2}$$

with $\theta = m_\eta(f_E(g(\mathcal{D}_N)))$ and the indexing $[[c]]$ for $c \in \{0, 1\}$ indicates the corresponding predicted class probabilities. Note that the parameters of the complex model $f$ do not undergo any modifications during the fine-tuning phase since this is a form of parameter-efficient fine-tuning.

**Phase 2: Additional fine-tuning of the MLP.** The distilled MLP $h_\theta$ is extracted by prompting $f_E$ with the same training dataset $\mathcal{D}_N$. One can further fine-tune $h_\theta$ for $K$ additional epochs on $\mathcal{D}_N$. During inference, the predictions are made using $h_\theta$, similar to any ordinary MLP. The complex model $f$ is no longer involved in the inference phase after the initial extraction of $h_\theta$.

**Nature of the input prompt:** The same small training set $\mathcal{D}_N$ (or a subset of $\mathcal{D}_N$) is used for two tasks during Phase 1: First, a serialized/transformed version of $\mathcal{D}_N$ (i.e., $g(\mathcal{D}_N)$) is used to prompt the base model $f$ to retrieve $h_\theta$. Then, $\mathcal{D}_N$ is used separately again (without serialization) to compute the cross-entropy loss of $h_\theta$, i.e., $\mathcal{L}(\eta, \mathcal{D}_N)$. Accordingly, we re-arrange the training set $\mathcal{D}_N$ to the following serialized/transformed structure for prompting (an example from the Calhousing dataset):

---

**Prompt (N=4):**

Example 0: The median income is 4.3292. The housing median age is 14.0. The total rooms is 4412.0. The total number of bedrooms is 952.0. The population is 1656.0. The number of households is 874.0. The latitude is 33.77. The longitude is -117.84. Is this house block valuable? Yes or no? The answer is yes.

Example 1: The median income is 3.7813. The housing median age is 41.0. The total rooms is 3170.0. The total number of bedrooms is 622.0. The population is 1091.0. The number of households is 528.0. The latitude is 37.9. The longitude is -122.54. Is this house block valuable? Yes or no? The answer is yes.

Example 2: The median income is 3.2731. The housing median age is 20.0. The total rooms is 5998.0. The total number of bedrooms is 1320.0. The population is 3185.0. The number of households is 1199.0. The latitude is 33.93. The longitude is -117.45. Is this house block valuable? Yes or no? The answer is no.

Example 3: The median income is 1.6955. The housing median age is 24.0. The total rooms is 2316.0. The total number of bedrooms is 599.0. The population is 1829.0. The number of households is 532.0. The latitude is 34.0. The longitude is -117.4. Is this house block valuable? Yes or no? The answer is no.

---

For computing the cross-entropy loss using $\mathcal{D}_N$, we let `X_train` denote an $N \times d$ tensor which includes the normalized feature values corresponding to the same $N$ examples in the `prompt`. These feature vectors are used as the input to $h_\theta$ for computing $\mathcal{L}(\eta, \mathcal{D}_N)$. `y_train` denotes the corresponding labels, also used for computing $\mathcal{L}(\eta, \mathcal{D}_N)$. In this example, the datapoints used for creating both the `prompt` and the `X/y_train` are the same and $N = 4$. However, when $N$ is sufficiently large, we may use different subsets of datapoints from $\mathcal{D}_N$ to create the `prompt` and `X/y_train`. Moreover, if $N$ is even larger, we may generate multiple examples of the above form with non-overlapping subsets from $\mathcal{D}_N$. Notice that when the base model is TabPFN, we directly use $\mathcal{D}_N$ for fine-tuning the transformer model without serialization.

The few-shot regime poses the inherent problem of **overfitting**. To overcome this problem, in each epoch, we randomly permute the feature order of $\mathcal{D}_N$. E.g., if in epoch 1 the order was (`age,`

education, label), in epoch 2 it will be (education, age, label). All examples in a prompt will have the same feature order. $\mathcal{D}_N$ is chosen to be class-wise balanced. A validation accuracy is computed on the same set $\mathcal{D}_N$ with a different randomly permuted feature order and is used for determining the hyperparameters such as the number of epochs and the complexity of $h_\theta$.

See Figure 2 for an illustration of the framework. Algorithm 1 summarizes the procedure. See Appendix A for more details on the exact training parameters corresponding to each dataset and $N$.

---

**Algorithm 1:** TabDistill framework

---

**Input:** Few-shot dataset $\mathcal{D}_N$, complex model $f$ with encoder $f_E$, transform $g(\cdot)$, number of fine-tuning epochs $T$, number of post-fine-tuning epochs $K$, architecture of the MLP $h_\theta$
**Output:** Trained MLP $h_\theta$
PHASE 1: Fine-tuning the transformer model to get a good output MLP
Initialize the linear mapping $m_\eta(z) : \mathcal{Z} \to \Theta$ based on the architecture of $h_\theta$ ;
**for** $t \leftarrow 1$ **to** $T$ **do**
    Randomly permute the feature order of $\mathcal{D}_N$ ;
    Create subsets $D_s, D_q \subseteq \mathcal{D}_N$;
    Generate Prompt $\leftarrow g(D_s)$, X_train $\leftarrow \{x : (x,y) \in D_q\}$ and
    y_train $\leftarrow \{y : (x,y) \in D_q\}$;
    Prompt the base model $f$ and obtain embeddings $z \leftarrow f_E(\text{Prompt})$ ;
    Infer parameters $\theta \leftarrow m_\eta(z)$ for the MLP $h_\theta$;
    Compute cross-entropy loss of classifier $h_\theta$: $\mathcal{L}(\eta, D_q)$, as in equation 2 with
    $x_n \in$ X_train and $y_n \in$ y_train;
    Update $\eta$ using gradient descent with gradients $\nabla_\eta \mathcal{L}(\eta, D_q)$
**end**
Prompt the fine-tuned base model $f$ with original dataset $\mathcal{D}_N$ and obtain the output MLP $h_\theta$;
PHASE 2: Additional fine-tuning of the obtained MLP if desired
**for** $k \leftarrow 1$ **to** $K$ **do**
    Compute cross-entropy loss $\mathcal{J}(\theta, D_q)$ with $x_n \in$ X_train and $y_n \in$ y_train;
    Update $\theta$ using gradient descent with gradients $\nabla_\theta \mathcal{J}(\theta, D_q)$ ;
**end**
**return** $h_\theta$ ;

---

### 2.3 PROPOSED INSTANTIATIONS OF TABDISTILL WITH TABPFN AND T0PP

**TabDistill with TabPFN:** TabPFN (Hollmann et al., 2023) is a transformer-based model pre-trained on a large number of synthetic tabular datasets. Tabular data (after pre-processing steps such as normalizing and one-hot encoding) can directly be used as the input to the TabPFN model. Therefore, the transform $g(x)$ in this case is the identity function. The TabPFN library provides a scikit-learn-style fit and predict functionality. In each training epoch, we fit the TabPFN classifier to $\mathcal{D}_N$ (with a randomly-permuted feature order) and obtain $z = f_E(\mathcal{D}_N)$. Next, we get $\theta = m_\eta(z)$ and compute the loss $\mathcal{L}(\eta; \mathcal{D}_N)$ in equation 2 to perform a gradient descent update on $\eta$. At the end of the training phase, $h_\theta$ is obtained by inputting $\mathcal{D}_N$ to TabPFN encoder without any permutations to the feature order. Finally, $h_\theta$ is fine-tuned on $\mathcal{D}_N$ for additional $K = 100$ epochs. The encoder output dimensionality dim($\mathcal{Z}$) of TabPFN varies with the number of training examples in multiples of 192. Consequently, the dimensionality of the matrix $A$ in the mapping $m_\eta(z)$ is taken to be dim($\Theta$) $\times$ 192$N$.

**TabDistill with T0pp:** The BigScience T0pp (Sanh et al., 2021) is an encoder-decoder style LLM trained on a large number of English language tasks specified in natural language prompts. This model has been used as the base LLM for TabLLM (Hegselmann et al., 2023). Since the input to the model has to be a natural language prompt, we convert the training data $\mathcal{D}_N$ (or a subset) into natural language using the "The <column_name> is <value>" style text template. $g(x)$ in this case represents this transform from tabular data to a natural language prompt. See Figure 1 for a detailed illustration. Appendix B lists example serializations used for each dataset. The training and inference phases are similar to that of TabDistill with TabPFN. In the end, the resultant MLP $h_\theta$ is fine-tuned on $\mathcal{D}_N$ for additional $K = 100$ epochs. The dimensionality of the encoder output $z = f_E(g(\mathcal{D}_N))$ is 4096. Therefore, the dimensionality of the matrix $A$ in the mapping $m_\eta(z)$ is taken to be dim($\Theta$) $\times$ 4096.

# 3 EXPERIMENTAL RESULTS

**Datasets and metrics:** We evaluate the TabDistill framework on four publicly available tabular datasets: *Bank* (UCI Bank Marketing) (Moro et al., 2014), *Blood* (UCI Blood Transfusion Service Center) (Yeh, 2008), *Calhousing* (California Housing Prices) (Pace & Barry, 1997) and *Income* (Census Income) (Kohavi, 1996). We divide each dataset into a train and a test split. $\mathcal{D}_N$ is selected from the training split. More details about the datasets are given in Table 5. Performance of all the models is compared with respect to the Receiver Operating Characteristic Area Under the Curve (ROC-AUC) metric. We consider the few-shot regime where the number of training examples is very low, specifically, $N \in \{4, 8, 16, 32, 64\}$.

**Baselines and $h_\theta$:** The architecture of $h_\theta$ is constant across all the experiments, unless specified explicitly. $h_\theta$ consisted of two hidden layers (hence, four layers in total, i.e., $R = 4$) with 10 neurons each (i.e., $L = 10$). We compare TabDistill with 3 simple and efficient classical baselines: logistic regression, XGBoost (Chen & Guestrin, 2016), an MLP with an architecture similar to $h_\theta$ but trained independently. In addition, we provide a performance comparison w.r.t. the base models TabPFN (Hollmann et al., 2023) and T0pp (Hegselmann et al., 2023) for completeness. All the models use the same set of labeled examples as TabDistill for training. Logistic regression and XGBoost are the best performing classical models in (Hegselmann et al., 2023), and hence, provide a strong baseline. The performance of the independently trained MLP helps observing the performance improvement obtained as a result of the distillation process.

**Hyperparameters:** Hyperparameters of all the baselines except were tuned using 4-fold cross-validation similar to Hegselmann et al. (2023), except in the case of training set size 4. When the training set size is 4, 2-fold cross-validation was used. We use `Scikit-learn`'s `GridSearchCV` and `RandomizedSearchCV` for tuning the hyperparameters. For XGBoost and MLP, we adopt the hyperparameter search ranges given in Grinsztajn et al. (2022). However, we keep the architecture of the MLP fixed to that of $h_\theta$. See Table 6 for more details on hyperparameter tuning of the baselines. TabPFN does not require any hyperparameters to be tuned (Hollmann et al., 2023). `Weights and Biases` sweeps were used for optimizing the hyperparameters of TabDistill, based on a validation score computed using the same training set $\mathcal{D}_N$. See Figure 4 in Appendix A for an example sweep.

**Main observations:** Table 1 presents the ROC-AUC of TabDistill along with that of the baselines, over the four tabular datasets. TabDistill shows superior performance over its classical counterparts particularly in the very few-shot regime. In general, the performance increases with the number of labeled examples available (i.e., with increasing $N$). Out of the three classical baselines, none seems to be universally better in performance across the datasets or the number of labeled examples. TabDistill + TabPFN shows better performance that TabDistill+T0pp in most cases, except in the Income dataset, where TabDistill+T0pp performs consistently better.

**Effect of the complexity of $h_\theta$:** In Table 2 we study the effect of the complexity of $h_\theta$ measured in terms of the number of layers $R$. The layer size $L$ is kept constant at 10. Bank dataset and TabPFN base model were used for the evaluation. As it is evident from the results, when the complexity of $h_\theta$ increases beyond a certain limit, the performance degrades.

**Performance with respect to the base models:** Table 3 presents the performance of the MLP $h_\theta$ obtained using TabDistill compared to the corresponding transformer-based model $f$ used for distillation. Interestingly, in some cases, the MLP $h_\theta$ distilled using our method surpasses the performance of the base model $f$ which it was distilled from.

**Feature attribution comparison:** We compute the Shapley feature attribution scores (Shapley et al., 1953) using the `SHAP` library for the classical baseline models logistic regression and XGBoost, and $h_\theta$ using the Calhousing dataset. The number of training examples used was 16 and the base model $f$ was TabPFN. Figure 3 shows the corresponding beeswarm plots for each baseline. We observe that the `median income` and the `longitude` have a greater impact on the output across all the models, indicating that the distilled models are consistent with the baselines trained in the ordinary fashion (See figures 3a, 3b and 3c). We also compute the attributions scores corresponding to an MLP distilled using a $\mathcal{D}_N$ with feature columns permuted (Figure 3d). Despite the permutation, this model displays feature importances similar to the original $h_\theta$. Hence, it is evident that the base model has correctly identified the correlation between the MLP weights and the feature order.

Table 1: Test ROC-AUC performance of TabDistill compared with the classical baselines. Best performance corresponding to each $N$ and dataset is emphasized in **bold**. Reported values are the average of 5 runs with different random states. The standard deviations are given as subscripts.

| Dataset | Method | Number of labeled examples ($N$) | | | | |
|---|---|---|---|---|---|---|
| | | 4 | 8 | 16 | 32 | 64 |
| Bank | MLP | $0.57_{.08}$ | $0.61_{.11}$ | $\mathbf{0.72}_{\mathbf{.05}}$ | $0.76_{.04}$ | $\mathbf{0.81}_{\mathbf{.03}}$ |
| | Logistic Regression | $0.54_{.10}$ | $0.65_{.06}$ | $\mathbf{0.72}_{\mathbf{.03}}$ | $0.73_{.04}$ | $0.77_{.04}$ |
| | XGBoost | $0.50_{.00}$ | $0.56_{.10}$ | $0.72_{.08}$ | $0.78_{.04}$ | $0.81_{.02}$ |
| | TabDistill + TabPFN (ours) | $\mathbf{0.72}_{\mathbf{.01}}$ | $0.67_{.06}$ | $0.68_{.02}$ | $\mathbf{0.79}_{\mathbf{.02}}$ | $0.81_{.02}$ |
| | TabDistill + T0pp (ours) | $0.70_{.02}$ | $\mathbf{0.67}_{\mathbf{.02}}$ | $0.72_{.01}$ | $0.74_{.02}$ | $0.80_{.02}$ |
| Blood | MLP | $0.57_{.10}$ | $0.61_{.09}$ | $0.60_{.09}$ | $0.61_{.07}$ | $0.67_{.08}$ |
| | Logistic Regression | $0.60_{.16}$ | $0.66_{.12}$ | $0.63_{.11}$ | $0.65_{.10}$ | $0.73_{.03}$ |
| | XGBoost | $0.50_{.00}$ | $0.55_{.09}$ | $0.55_{.07}$ | $0.65_{.07}$ | $0.72_{.02}$ |
| | TabDistill + TabPFN (ours) | $0.56_{.07}$ | $\mathbf{0.67}_{\mathbf{.05}}$ | $\mathbf{0.69}_{\mathbf{.07}}$ | $\mathbf{0.68}_{\mathbf{.09}}$ | $\mathbf{0.75}_{\mathbf{.00}}$ |
| | TabDistill + T0pp (ours) | $\mathbf{0.62}_{\mathbf{.08}}$ | $0.58_{.08}$ | $0.67_{.06}$ | $0.67_{.04}$ | $0.68_{.06}$ |
| Calhousing | MLP | $0.49_{.07}$ | $0.63_{.10}$ | $0.72_{0.12}$ | $0.79_{.07}$ | $0.82_{.04}$ |
| | Logistic Regression | $0.59_{.10}$ | $0.66_{.13}$ | $0.74_{.14}$ | $\mathbf{0.83}_{\mathbf{.04}}$ | $\mathbf{0.89}_{\mathbf{.01}}$ |
| | XGBoost | $0.50_{.00}$ | $0.57_{.10}$ | $\mathbf{0.75}_{\mathbf{.04}}$ | $0.75_{.06}$ | $0.81_{.06}$ |
| | TabDistill + TabPFN (ours) | $0.64_{.06}$ | $0.65_{.03}$ | $0.65_{.03}$ | $0.77_{.03}$ | $0.84_{.00}$ |
| | TabDistill + T0pp (ours) | $\mathbf{0.67}_{\mathbf{.05}}$ | $\mathbf{0.67}_{\mathbf{.03}}$ | $0.66_{.05}$ | $0.74_{.03}$ | $0.81_{.01}$ |
| Income | MLP | $0.51_{.10}$ | $0.69_{.05}$ | $0.74_{.07}$ | $0.78_{.04}$ | $0.79_{.04}$ |
| | Logistic Regression | $\mathbf{0.76}_{\mathbf{.07}}$ | $0.75_{.09}$ | $0.79_{.02}$ | $0.82_{.02}$ | $0.84_{.03}$ |
| | XGBoost | $0.50_{.00}$ | $0.57_{.11}$ | $0.65_{.14}$ | $0.80_{.02}$ | $0.81_{.01}$ |
| | TabDistill + TabPFN (ours) | $0.68_{.08}$ | $0.75_{.03}$ | $0.80_{.02}$ | $0.81_{.02}$ | $0.83_{.01}$ |
| | TabDistill + T0pp (ours) | $0.70_{.03}$ | $\mathbf{0.77}_{\mathbf{.02}}$ | $\mathbf{0.83}_{\mathbf{.01}}$ | $\mathbf{0.83}_{\mathbf{.02}}$ | $\mathbf{0.85}_{\mathbf{.01}}$ |

Table 2: Test ROC-AUC performance of TabDistill with different MLP complexities

| # Labeled examples | Number of layers (R) | | | |
|---|---|---|---|---|
| | 2 | 4 | 8 | 16 |
| 4 | $0.72_{.02}$ | $0.72_{.01}$ | $0.65_{.09}$ | $0.53_{.04}$ |
| 8 | $0.74_{.01}$ | $0.67_{.06}$ | $0.72_{.03}$ | $0.50_{.00}$ |

## 4 CONCLUSION

We introduce TabDistill, a novel distillation framework for extracting the pre-trained knowledge of transformer models into neural networks for classifying tabular data. The framework produces MLPs with enhanced performance particularly when the labeled data is limited. Experiments show that the resulting MLPs surpass the classical machine learning models such as XGBoost and logistic regression, and in some cases, the initial transformer model used for distilling itself in the few-shot regime. TabDistill can be used to generate scalable, computationally efficient models with a small number of training data, bringing together the advantages of transformers and classical models.

**Limitations and future directions:** While TabDistill produces MLPs which surpass the classical models in the few-shot regime, the performance gain is limited when the number of labeled examples increase. Hence, there is room for improvement when the training set is large. The linear mapping function $m_\eta(\cdot)$ used in the current experiments can be replaced with other alternatives to potentially achieve performance improvements. Moreover, the extracted MLP may inherit the biases of the base model, although it can be mitigated up to some extent through the MLP finetuning in the second phase.

Table 3: Test ROC-AUC performance of TabDistill compared with the base model $f$. Best performance corresponding to each $N$ and dataset is emphasized in **bold**. Reported values are the average of 5 runs with different random states. The standard deviations are given as subscripts.

| Dataset | Method | Number of labeled examples ($N$) | | | | |
|---|---|---|---|---|---|---|
| | | 4 | 8 | 16 | 32 | 64 |
| Bank | TabPFN (11M params) | $0.62_{.05}$ | $\mathbf{0.68_{.08}}$ | $\mathbf{0.75_{.08}}$ | $\mathbf{0.82_{.05}}$ | $\mathbf{0.86_{.02}}$ |
| | TabDistill + TabPFN (ours) | $\mathbf{0.72_{.01}}$ | $0.67_{.06}$ | $0.68_{.02}$ | $0.79_{.02}$ | $0.81_{.02}$ |
| | T0pp (TabLLM, 11B params)[†] | $0.59_{.10}$ | $0.64_{.05}$ | $0.65_{.05}$ | $0.64_{0.6}$ | $0.69_{.03}$ |
| | TabDistill + T0pp (ours) | $\mathbf{0.70_{.02}}$ | $\mathbf{0.67_{.02}}$ | $\mathbf{0.72_{.01}}$ | $\mathbf{0.74_{.02}}$ | $\mathbf{0.80_{.02}}$ |
| Blood | TabPFN (11M params) | $0.55_{.20}$ | $0.61_{.14}$ | $0.59_{.12}$ | $\mathbf{0.68_{.07}}$ | $0.73_{.02}$ |
| | TabDistill + TabPFN (ours) | $\mathbf{0.56_{.07}}$ | $\mathbf{0.67_{.05}}$ | $\mathbf{0.69_{.07}}$ | $0.68_{.09}$ | $\mathbf{0.75_{.00}}$ |
| | T0pp (TabLLM, 11B params)[†] | $0.58_{.09}$ | $\mathbf{0.66_{.03}}$ | $0.66_{.07}$ | $\mathbf{0.68_{.04}}$ | $\mathbf{0.68_{.04}}$ |
| | TabDistill + T0pp (ours) | $\mathbf{0.62_{.08}}$ | $0.58_{.08}$ | $\mathbf{0.67_{.06}}$ | $0.67_{.04}$ | $\mathbf{0.68_{.06}}$ |
| Calhousing | TabPFN (11M params) | $0.59_{.08}$ | $\mathbf{0.70_{.10}}$ | $\mathbf{0.83_{.04}}$ | $\mathbf{0.84_{.04}}$ | $\mathbf{0.88_{.02}}$ |
| | TabDistill + TabPFN (ours) | $\mathbf{0.64_{.06}}$ | $0.65_{.03}$ | $0.65_{.03}$ | $0.77_{.03}$ | $0.84_{.00}$ |
| | T0pp (TabLLM, 11B params)[†] | $0.63_{.05}$ | $0.60_{.07}$ | $\mathbf{0.70_{.08}}$ | $\mathbf{0.77_{.08}}$ | $0.77_{0.4}$ |
| | TabDistill + T0pp (ours) | $\mathbf{0.67_{.05}}$ | $\mathbf{0.67_{.03}}$ | $0.66_{.05}$ | $0.74_{.03}$ | $\mathbf{0.81_{.01}}$ |
| Income | TabPFN (11M params) | $\mathbf{0.69_{.06}}$ | $0.74_{.09}$ | $0.78_{.01}$ | $\mathbf{0.82_{.03}}$ | $\mathbf{0.84_{.01}}$ |
| | TabDistill + TabPFN (ours) | $0.68_{.08}$ | $\mathbf{0.75_{.03}}$ | $\mathbf{0.80_{.02}}$ | $0.81_{.02}$ | $0.83_{.01}$ |
| | T0pp (TabLLM, 11B params)[†] | $\mathbf{0.84_{.01}}$ | $\mathbf{0.84_{.02}}$ | $\mathbf{0.84_{.04}}$ | $\mathbf{0.84_{.01}}$ | $0.84_{.02}$ |
| | TabDistill + T0pp (ours) | $0.70_{.03}$ | $0.77_{.02}$ | $0.83_{.01}$ | $0.83_{.02}$ | $\mathbf{0.85_{.01}}$ |

[†] TabLLM performance values are as reported in Hegselmann et al. (2023)

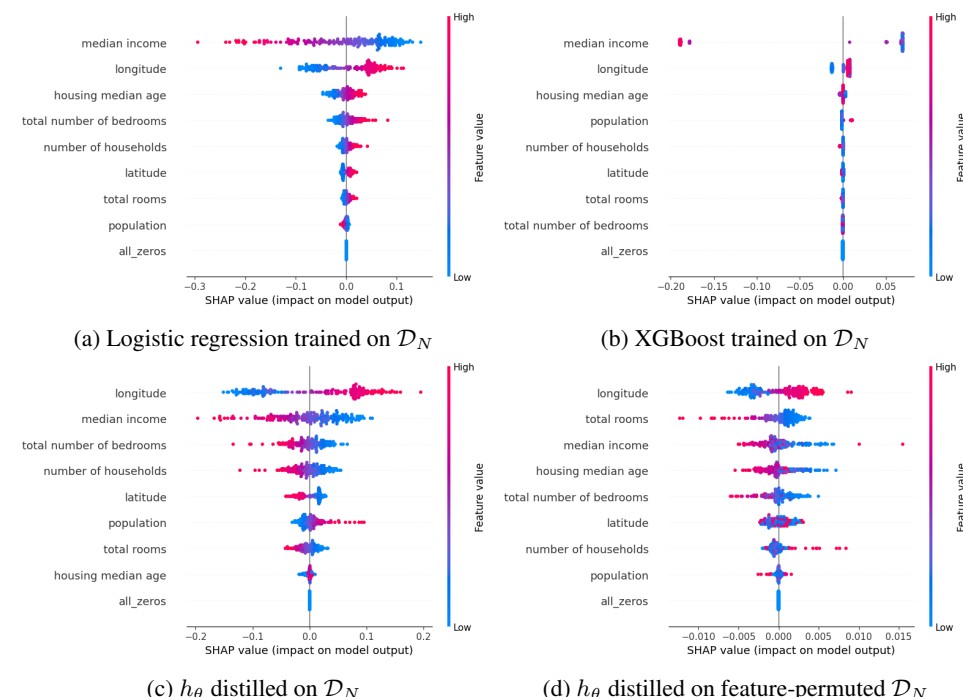

(a) Logistic regression trained on $\mathcal{D}_N$      (b) XGBoost trained on $\mathcal{D}_N$

(c) $h_\theta$ distilled on $\mathcal{D}_N$      (d) $h_\theta$ distilled on feature-permuted $\mathcal{D}_N$

Figure 3: SHAP feature attributions. Computed on the Calhousing dataset with TabPFN as the base model $f$. Training set size $N$ is 16. 200 samples were used for computing the beeswarm plots.

## 5 REPRODUCIBILITY STATEMENT

The TabDistill framework has been explained in detail under Section 2. Details on the experimental setup including the datasets and the baselines are given in Section 3. Further details including hyperparameters and training setup are given in Appendix A. All the experiments were done on computer with a 3.5 GHz AMD EPYC 7763 64-Core Processor and an Nvidia RTX 6000 Ada GPU.

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

## A  TRAINING SETUP AND HYPERPARAMETER OPTIMIZATION

The PHASE 1 fine-tuning was carried out for 300 epochs. The epoch with the best validation accuracy (computed using a randomly permuted version of the $\mathcal{D}_N$) was used for inferring the weights for the final $h_\theta$. Learning rates were selected from the set $[1e^{-6}, 2e^{-4}]$. Adam optimizer was used with a weight decay of $1e^{-3}$. For some of the experiments with $N = 4$, weight decay was set to 0. The subsets $D_s$ and $D_q$ of the training set $\mathcal{D}_N$ were selected as per Table 4. More details on the datasets including the test-train split sizes is given in Table 5.

Table 4: The scheme of partitioning $\mathcal{D}_N$

| Parameter | Number of labeled examples | | | | |
|---|---|---|---|---|---|
| | 4 | 8 | 16 | 32 | 64 |
| $\lvert D_s \rvert$ | 4 | 4 | 8 | 8 | 8 |
| $\lvert D_q \rvert$ | 4 | 4 | 8 | 8 | 8 |
| $D_s = D_q$? | True | True | False | False | False |
| # of $(D_s, D_q)$ pairs | 1 | 2 | 1 | 2 | 4 |

Table 5: Dataset details

| Dataset | # Features | Test size | Train size | Target |
|---|---|---|---|---|
| Bank | 16 | 43211 | 2000 | To predict whether the client will subscribe a term deposit |
| Blood | 4 | 374 | 374 | To predict whether a person would donate blood |
| Calhousing | 12 | 19640 | 1000 | To predict whether a given house block is valuable or not |
| Income | 12 | 44222 | 1000 | To predict whether a person's annual income exceeds 50K |

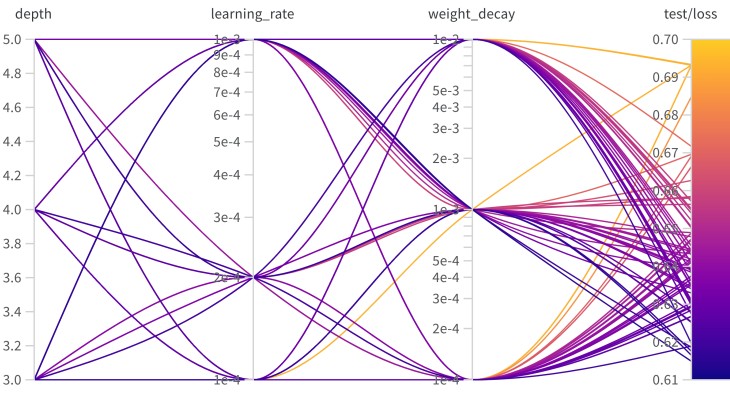

Figure 4: `Weights and Biases` sweeps used for optimizing hyperparameters for TabDistill with TabPFN and Calhousing dataset, 64 training examples

Hyperparameter optimization for the classical baseline models was done using `scikit-learn`'s `GridSearchCV` and `RandomizedSearchCV` methods. The search ranges are given in Table 6. Hyperparameters for the TabDistill framework were deteremined using `Weights and Biases` sweeps. Figure 4 illustrates one such sweep corresponding to the TabPFN base model and the Calhousing dataset.

Table 6: Hyperparameter ranges used for classical baseline models

| Model | Hyperparameter | Range/Distribution | Method |
|---|---|---|---|
| MLP | Number of layers
Hidden layer size
Number of epochs
Learning rate | 4
10
[30, 50, 100, 300]
[1e-5, 1e-4, 1e-3, 1e-2] | Grid search |
| Logistic Regression | C | [0.01, 0.1, 1, 10] | Grid search |
| XGBoost | Max depth
Number of estimators
Min child weight
Subsample
Learning rate
Column sample by level
Column sample by tree
Gamma
Lambda
Alpha | UniformInt[1,11]
1000
LogUniformInt[1, 1e2]
Uniform[0.5, 1]
LogUniform[1e-5, 0.7]
Uniform[0.5, 1]
Uniform[0.5, 1]
LogUniform[1e-8, 7]
LogUniform[1, 4]
LogUniform[1e-8, 1e2] | Randomized search with 20 iterations |

## B    SERIALIZATIONS USED FOR PROMPTING T0PP

**Bank dataset, N=4:**

Example 0: The age is 29.0. The job is blue-collar. The marital status is married. The education is secondary. The default is no. The account balance is 314.0. The housing loan is available. The personal loan is available. The contact communication type is cellular. The last contact day of the month is 17.0. The last contact month of year is apr. The last contact duration, in seconds is 357.0. The number of contacts in campaign is 1.0. The days since last contact is -1.0. The number of previous contacts is 0.0. The previous contact outcome is unknown. Does this client subscribe to a term deposit? Yes or no? The answer is no.

Example 1: The age is 62.0. The job is housemaid. The marital status is married. The education is unknown. The default is no. The account balance is 2021.0. The housing loan is not available. The personal loan is not available. The contact communication type is telephone. The last contact day of the month is 26.0. The last contact month of year is feb. The last contact duration, in seconds is 361.0. The number of contacts in campaign is 1.0. The days since last contact is -1.0. The number of previous contacts is 0.0. The previous contact outcome is unknown. Does this client subscribe to a term deposit? Yes or no? The answer is yes.

Example 2: The age is 32.0. The job is blue-collar. The marital status is single. The education is secondary. The default is no. The account balance is 3.0. The housing loan is available. The personal loan is not available. The contact communication type is unknown. The last contact day of the month is 23.0. The last contact month of year is may. The last contact duration, in seconds is 108.0. The number of contacts in campaign is 3.0. The days since last contact is -1.0. The number of previous contacts is 0.0. The previous contact outcome is unknown. Does this client subscribe to a term deposit? Yes or no? The answer is no.

Example 3: The age is 36.0. The job is management. The marital status is married. The education is tertiary. The default is no. The account balance is 203.0. The housing loan is not available. The personal loan is not available. The contact communication type is cellular. The last contact day of the month is 25.0. The last contact month of year is jan. The last contact duration, in seconds is 255.0. The number of contacts in campaign is 1.0. The days since last contact is 88.0. The number of previous contacts is 1.0. The previous contact outcome is success. Does this client subscribe to a term deposit? Yes or no? The answer is yes.

**Blood dataset, N=4:**
Example 0: The previous blood donation record of a person is as follows: The months since last donation is 14.0. The total number of donations is 3.0. The total amount of blood donated (in cc) is 750.0. The months since first donation is 21.0. Will this person donate blood next time? Yes or no? The answer is no.

Example 1: The previous blood donation record of a person is as follows: The months since last donation is 4.0. The total number of donations is 2.0. The total amount of blood donated (in cc) is 500.0. The months since first donation is 4.0. Will this person donate blood next time? Yes or no? The answer is yes.

Example 2: The previous blood donation record of a person is as follows: The months since last donation is 16.0. The total number of donations is 7.0. The total amount of blood donated (in cc) is 1750.0. The months since first donation is 87.0. Will this person donate blood next time? Yes or no? The answer is yes.

Example 3: The previous blood donation record of a person is as follows: The months since last donation is 11.0. The total number of donations is 5.0. The total amount of blood donated (in cc) is 1250.0. The months since first donation is 35.0. Will this person donate blood next time? Yes or no? The answer is no.

**Calhousing dataset, N=4:**
Example 0: The median income is 4.3292. The housing median age is 14.0. The total rooms is 4412.0. The total number of bedrooms is 952.0. The population is 1656.0. The number of households is 874.0. The latitude is 33.77. The longitude is -117.84. Is this house block valuable? Yes or no? The answer is yes.

Example 1: The median income is 3.7813. The housing median age is 41.0. The total rooms is 3170.0. The total number of bedrooms is 622.0. The population is 1091.0. The number of households is 528.0. The latitude is 37.9. The longitude is -122.54. Is this house block valuable? Yes or no? The answer is yes.

Example 2: The median income is 3.2731. The housing median age is 20.0. The total rooms is 5998.0. The total number of bedrooms is 1320.0. The population is 3185.0. The number of households is 1199.0. The latitude is 33.93. The longitude is -117.45. Is this house block valuable? Yes or no? The answer is no.

Example 3: The median income is 1.6955. The housing median age is 24.0. The total rooms is 2316.0. The total number of bedrooms is 599.0. The population is 1829.0. The number of households is 532.0. The latitude is 34.0. The longitude is -117.4. Is this house block valuable? Yes or no? The answer is no.

**Income dataset, N=4:**
Example 0: The age is 35. The workclass is Private. The education is HS-grad. The marital-status is Married-civ-spouse. The occupation is Transport-moving. The relationship is Husband. The race is White. The sex is Male. The capital-gain is 0. The capital-loss is 0. The hours-per-week is 30. The native-country is United-States. Does this person make over 50K a year? Answer with Yes or No. The answer is No.

Example 1: The age is 32. The workclass is Self-emp-not-inc. The education is 10th. The marital-status is Married-civ-spouse. The occupation is Exec-managerial. The relationship is Husband. The race is White. The sex is Male. The capital-gain is 0. The capital-loss is 0. The hours-per-week is 55. The native-country is United-States. Does this person make over 50K a year? Answer with Yes or No. The answer is No.

Example 2: The age is 56. The workclass is Private. The education is Some-college. The marital-status is Married-civ-spouse. The occupation is Sales. The relationship is Husband. The race is White. The sex is Male. The capital-gain is 0. The capital-loss is 0. The hours-per-week is 45. The native-country is United-States. Does this person make over 50K a year? Answer with Yes or No. The answer is Yes.

Example 3: The age is 44. The workclass is State-gov. The education is Masters. The marital-status is Married-civ-spouse. The occupation is Prof-specialty. The relationship is Husband. The race is White. The sex is Male. The capital-gain is 7688. The capital-loss is 0. The hours-per-week is 50. The native-country is United-States. Does this person make over 50K a year? Answer with Yes or No. The answer is Yes.

## C  ACRONYMS

**GBDT** Gradient Boosted Decision Tree. 1, 2

**LLM** Large Language Model. 2–4, 6

**MLP** Multi-Layer Perceptron. 2–8, 13

**ROC-AUC** Receiver Operating Characteristic Area Under the Curve. 7

