# OpenReview forum: "Distilling Transformers into Neural Nets for Few-Shot Tabular Classification"
_ICLR.cc/2026/Conference — ICLR 2026 Conference Withdrawn Submission_

### Official Review · Reviewer_e6Xc · 2025-10-24

**Soundness:** 1
**Presentation:** 3
**Contribution:** 1
**Rating:** 2
**Confidence:** 5

**Summary:**

The paper presents a way of distilling a tabular transformers to MLP for few-shot learning task. The authors want to reduce the complexity of transformers by using simpler, more effective architecture. In fact, the authors use transformer as a hypernetwork, which produces the weights to target network (MLP) for solving target task. The method is evaluated on 4 datasets and compared with 3 baselines.

**Strengths:**

The authors consider unsolved few-shot learning problem for tabular data. They construct an interesting model using hypernetwork, which can be used even outside few-shot scenario.

**Weaknesses:**

As I mentioned, the authors construct a type of hypernetwork and therefore it is very similar approach to MotherNet (https://openreview.net/pdf?id=6H4jRWKFc3). The authors of MotherNet have different motivation, but the architecture is at first glance more or less the same. This reduces the novelty of the paper.

The experimental study is very limited and does not meet the standards of ICLR:
-4 datasets while current works use even hundred data
-3 baselines and none of them is designed to few-shot learning (see https://arxiv.org/abs/2303.00918 and https://proceedings.neurips.cc/paper_files/paper/2024/hash/40eff1670d6b08bb1bda48b0c5f30110-Abstract-Conference.html)
-no comparison with base transformers so we do not know what is the gain or lose

Transformers are complex but the use of TabPFN, which does not require retraining is straightforward. Therefore, I do not see the motivation of the paper especialy when the authors have to use base transformer to create MLP.

**Questions:**

The authors should elaborate about the connections of the proposed model and MotherNet. I could miss some important detail but  both approaches have the same architecture (they sightly differ at fine-tuning stage)

---

### Official Review · Reviewer_aQEb · 2025-10-30

**Soundness:** 2
**Presentation:** 3
**Contribution:** 2
**Rating:** 2
**Confidence:** 4

**Summary:**

This paper introduces TabDistill, a framework that transfers knowledge from pre-trained transformers to lightweight neural networks for few-shot tabular classification.

**Strengths:**

Strengths
	The paper presents a novel approach that transfers the representational power of pre-trained transformers into compact MLPs for few-shot tabular learning.
	The writing is clear, and the overall presentation is well structured and easy to follow.

**Weaknesses:**

Weakness
	Missing efficiency analysis:
Although the paper emphasizes parameter efficiency, it does not provide quantitative measurements of training time, inference time, or model size. Including a comparison table with these efficiency metrics would strengthen the claims.
	Limited dataset scale:
The experiments are conducted on only four small UCI datasets, which limits their generalizability. It is recommended to include larger or more challenging benchmarks—for example, datasets from TabLLM[1] or FeatLLM[2] for T0-based models, and classical tabular benchmarks such as the OpenML-CC18 suite or TabZilla[3] for TabPFN-based models.
	Insufficient baselines:
The comparison is limited to Logistic Regression, XGBoost, and MLP. It would be beneficial to include additional neural network baselines, such as SAINT[4] and SCARF[5]. Moreover, the original TabLLM[1], TabPFN[6], and TabPFNv2[7], could also be added to Table 1 to provide a more comprehensive comparison of performance across different models.
	No evaluation on multi-class datasets:
All current experiments are on binary classification. It is suggested to include at least one multi-class dataset to demonstrate scalability.
	Code not available:
The paper does not provide code. Code release would significantly improve transparency and verifiability.
[1] Tabllm: Few-shot classification of tabular data with large language models – aistats 2023
[2] Large Language Models Can Automatically Engineer Features for Few-Shot Tabular Learning-ICML 2024
[3] When Do Neural Nets Outperform Boosted Trees on Tabular Data-NeurIPS 2023
[4] SAINT: Improved Neural Networks for Tabular Data via Row Attention and Contrastive Pre-Training – NeurIPS 2022 workshop
[5] Scarf: Self-Supervised Contrastive Learning using Random Feature Corruption – ICLR 2022
[6] TabPFN: A Transformer That Solves Small Tabular Classification Problems in a Second – ICLR 2023
[7] Accurate predictions on small data with a tabular foundation model – nature 2025

**Questions:**

Questions:
	The MLP generated in Phase 1 (the mapping function training stage) can already make predictions, but its performance is not reported. Could the authors provide the ROC-AUC or accuracy of the Phase 1 model to clarify the contribution and effectiveness of each stage in the proposed framework?
	The proposed method learns a mapping function to generate MLP parameters instead of directly fine-tuning the MLP, which appears unnecessarily complex. What is the theoretical or empirical motivation behind this design? For instance, does it help prevent overfitting or improve generalization in few-shot scenarios?
	Details about the mapping m_η are needed. Is it simply a learnable matrix, or others?

---

### Official Review · Reviewer_cGjP · 2025-11-01

**Soundness:** 2
**Presentation:** 1
**Contribution:** 2
**Rating:** 2
**Confidence:** 2

**Summary:**

This paper proposes the distillation of a complex pre-trained transformer model into an MLP by fine-tuning the transformer to infer its weight. The fine-tuning is conducted on examples from tabular datasets, and the resulting model is used for downstream few-shot tabular classification.

**Strengths:**

I think the high-level idea of distilling knowledge from complex model to perform downstream tasks is interesting.

**Weaknesses:**

1. Clarity of the paper. I tried to read Section 2.1 and 2.2 a few times and I still don't quite understand how the MLP distillation process work. Either I have some reading deficiency, or the authors have not been doing a good job describing their method. Because although I don't fully understand, I can see that the proposed method is quite simple (not necessarily in a bad way) and heuristic, and it should not be too hard to describe clearly and it should not be too hard for the readers to understand. Here are a few suggestions: could you add dimensionalities to the definitions of notations such as $z$ and $f_E$? I'm not quite sure if it's a token-level representation, or aggregated representation for each example in the batch, or aggregated representation of the whole batch of examples. And could you more clearly describe how they are converted to each $W_i$ and $b_i$? I'm going to give my assessment a low confidence score for now.

2. My biggest problem with this work is its practicality. TabPFN by itself is able to achieve results much better than the ones reported in Table 1 by fitting way more examples in memory in one forward pass without the need for serialization, and it's probably much faster than the proposed method (if this is not true, please provide evidence) because it's a much smaller model and does not require fine-tuning. Therefore, the only scenario where the proposed method could be favored over the recent tabular foundation models such as TabPFN is if the practitioners are interested in tabular datasets that only have 10s of samples. Even if there are such datasets of interest, since there is no comparison to TabPFN directly, and no comparison to efficient fine-tuning of TabPFN such as TuneTables [1], There really isn't any fair comparison to show the proposed method's advantage in this scenario either.

3. More literature review & comparison to the most recent tabular foundation models such as TabPFN v2 [2], TabICL [3], APT [4] would strengthen the paper as well.

[1] Feuer, Benjamin, et al. "Tunetables: Context optimization for scalable prior-data fitted networks." Advances in Neural Information Processing Systems 37 (2024): 83430-83464.

[2] Hollmann, Noah, et al. "Accurate predictions on small data with a tabular foundation model." Nature 637.8045 (2025): 319-326.

[3] Qu, Jingang, et al. "TabICL: A Tabular Foundation Model for In-Context Learning on Large Data." CoRR (2025).

[4] Wu, Yulun, and Doron L. Bergman. "Zero-shot Meta-learning for Tabular Prediction Tasks with Adversarially Pre-trained Transformer." Forty-second International Conference on Machine Learning.

**Questions:**

Please see above.

---

### Official Review · Reviewer_qAG6 · 2025-11-03

**Soundness:** 2
**Presentation:** 2
**Contribution:** 1
**Rating:** 2
**Confidence:** 4

**Summary:**

This paper focuses on the few-shot tabular classification and proposes a distillation framework to distill various transformer based foundation models into MLPs via a two phased algorithm, by first finetuning a head on top of a foundation model that predicts small MLP parameters and then by finetuning said MLP with gradient descent on a given downstream task. The method is instantiated on top of TabPFN (seemingly v1) and T0pp models and evaluated on five tabular datasets adopted from prior research paper on extreme tabular few-shot classification.

**Strengths:**

The paper seems to be the first to try such an approach atop of an LLM-based tabular foundation model (not an ICL-based one) and it seems to work, which is novel and interesting.

**Weaknesses:**

First, the very important peace of related work is missed in [MotherNet](https://arxiv.org/abs/2312.08598). This work essentially does a very similar hyper-network like procedure with a PFN model, but during pre-training (which even amortizes the finetuning cost). Second, the evaluation is very limited and I am very skeptical of it's soundness. A total of five datasets is an outlier for todays tabular research (you can't make calls on such a small sample of datasets). And when we have only 4 to 64 samples, averaging resutls over multiple train data subsamples to ensure results are more reliable would make results more trustworthy. There are also baselines like [Tabula](https://proceedings.neurips.cc/paper_files/paper/2024/hash/4fd5cfd2e31bebbccfa5ffa354c04bdc-Abstract-Conference.html), CARTE, TabPFNv2, TabICL and LimiX missing (all these models are good candidates for the proposed method also). The actual results in Table 1 seems to be missing the original model performance for tabpfn and T0pp, and from the results in Table 1 it is not totally clear why would I prefer this method to linear regression - there are only two clear settings where it is not on par with the proposed methods (which in fairness may get eliminated if we try to average over different training subsets).

There are also minor issues like calling BERT an encoder-decoder model in line 57, or missing baselines after "except..." in line 344. This together with the larger issues from the first paragraph makes the paper feel rushed and in need of significant refinement.

**Questions:**

Some of the directions for improevment. I believe it is next to impossible to improve the current version with a short rebuttal and text diff.

- Discuss and compare against MotherNet baseline
- Use more modern tabular foundation models
- Use a more comprehensive benchmark (you can take the TabArena benchmark for example)
- Is there any insight on how and why the proposed method works? Is the finetuning step required? How it depends on foundation model quality?
- Why the focus on the ultra few shot niche? I think the general distillation formulation is far more useful and interesting in reality than 4-shot tabular classification.

---

### Note · Authors · 2025-11-25

**Comment:**

Dear Reviewers and the AC,

We would like to sincerely thank you for the time and effort you invested in evaluating our submission. Although we have decided to withdraw the paper, we greatly appreciate your thoughtful feedback and the care with which you approached the review process. Your insights will be invaluable as we refine the work further.

Thank you again for your dedication and for contributing to the quality of the review process.

**Withdrawal Confirmation:**

I have read and agree with the venue's withdrawal policy on behalf of myself and my co-authors.